# Accurate immune repertoire sequencing reveals malaria infection driven antibody lineage diversification in young children

Ben S. Wendel[1], Chenfeng He[2], Mingjuan Qu[2,7], Di Wu[2], Stefany M. Hernandez[1], Ke-Yue Ma[3], Eugene W. Liu[4,8], Jun Xiao[5], Peter D. Crompton[4], Susan K. Pierce[4], Pengyu Ren[2], Keke Chen[6] & Ning Jiang[2,3]

Accurately measuring antibody repertoire sequence composition in a small amount of blood is challenging yet important for understanding repertoire responses to infection and vaccination. We develop molecular identifier clustering-based immune repertoire sequencing (MIDCIRS) and use it to study age-related antibody repertoire development and diversification before and during acute malaria in infants (<12 months old) and toddlers (12–47 months old) with 4–8 ml of blood. Here, we show this accurate and high-coverage repertoire-sequencing method can use as few as 1000 naive B cells. Unexpectedly, we discover high levels of somatic hypermutation in infants as young as 3 months old. Antibody clonal lineage analysis reveals that somatic hypermutation levels are increased in both infants and toddlers upon infection, and memory B cells isolated from individuals who previously experienced malaria continue to induce somatic hypermutations upon malaria rechallenge. These results highlight the potential of antibody repertoire diversification in infants and toddlers.

[1] McKetta Department of Chemical Engineering, Cockrell School of Engineering, University of Texas at Austin, Austin, TX 78712, USA. [2] Department of Biomedical engineering, Cockrell School of Engineering, University of Texas at Austin, Austin, TX 78712, USA. [3] Institute for Cellular and Molecular Biology, College of Natural Sciences, University of Texas at Austin, Austin, TX 78712, USA. [4] Laboratory of Immunogenetics, National Institute of Allergy and Infectious Diseases, National Institutes of Health, Rockville, MD 20852, USA. [5] ImmuDX, LLC, Austin, TX 78750, USA. [6] Department of Computer Science and Engineering, Wright State University, Dayton, OH 45435, USA. [7] Present address: School of Life Sciences, Ludong University, Yantai, Shandong 264025, China. [8] Present address: Parasitic Diseases Branch, Division of Parasitic Diseases and Malaria, Center for Global Health, Atlanta 30329 GA, USA. Ben S. Wendel, Chenfeng He, and Mingjuan Qu contributed equally to this work. Correspondence and requests for materials should be addressed to N.J. (email: jiang@austin.utexas.edu)

V(D)J recombination creates hundreds of billions of antibodies and T cell receptors that collectively serve as the immune repertoire to protect the host from pathogens. Somatic hypermutation (SHM) further diversifies the antibody repertoire, which makes it impossible to quantify this diversity with nucleotide resolution until the development of high-throughput sequencing-based immune repertoire sequencing (IR-seq)[1–4]. Although we and others have developed methods to control for artifacts from high amplification bias and sequencing error rates through data analysis[3, 5–9], obtaining accurate sequencing information has now been made possible by the use of molecular identifiers (MID)[10–13]. MIDs serve as barcodes to track genes of interest through amplification and sequencing. They are short stretches of nucleotide sequence tags composed of randomized nucleotides that are usually tagged to cDNA during reverse transcription to identify sequencing reads that originated from the same mRNA transcript. Despite these advancements, the large amount of input RNA required and low diversity coverage make it challenging to analyze small numbers of cells, such as memory B cells from dissected tissues or blood draws from young children, using IR-seq because these samples require many PCR cycles to generate enough material to make sequencing libraries, thus exacerbating PCR bias and errors.

Here we report the development of MID clustering-based IR-seq (MIDCIRS) that further separates different RNA molecules tagged with the same MID. Using naive B cells, we demonstrate that MIDCIRS has a high coverage of the diversity estimate, or different types of antibody sequences, that is consistent with the input cell number and a large dynamic range of three orders of magnitude compared to other MID-based immune repertoire-sequencing methods[10, 11]. Given the wide use of IR-seq in basic research as well as clinical settings, we believe the method outlined here will serve as an important guideline for future IR-seq experimental designs.

As a proof of principle, we use MIDCIRS to examine the antibody repertoire diversification in infants (<12 months old) and toddlers (12–47 months old) from a malaria endemic region in Mali before and during acute *Plasmodium falciparum* infection. Although the antibody repertoire in fetuses[14], cord blood[15], young adults[6], and the elderly[6, 16] has been studied, infants and toddlers are among the most vulnerable age groups to many pathogenic challenges, yet their immune repertoires are not well understood. Infants are widely thought to have weaker responses than toddlers to vaccines because of their developing immune systems[17]. Thus, understanding how the antibody repertoire develops and diversifies during a natural infection, such as malaria, not only provides valuable insight into B cell ontology in humans, but also provides critical information for vaccine development for these two vulnerable age groups. Using peripheral blood mononuclear cells (PBMC) from 13 children aged 3–47 months old before and during acute malaria, with two of the children followed for a second year and nine additional pre-malaria individuals we show that infants and toddlers use the same V, D, and J combination frequencies and have similar complementarity determining region 3 (CDR3) length distributions. Although infants have a lower level of average SHM than

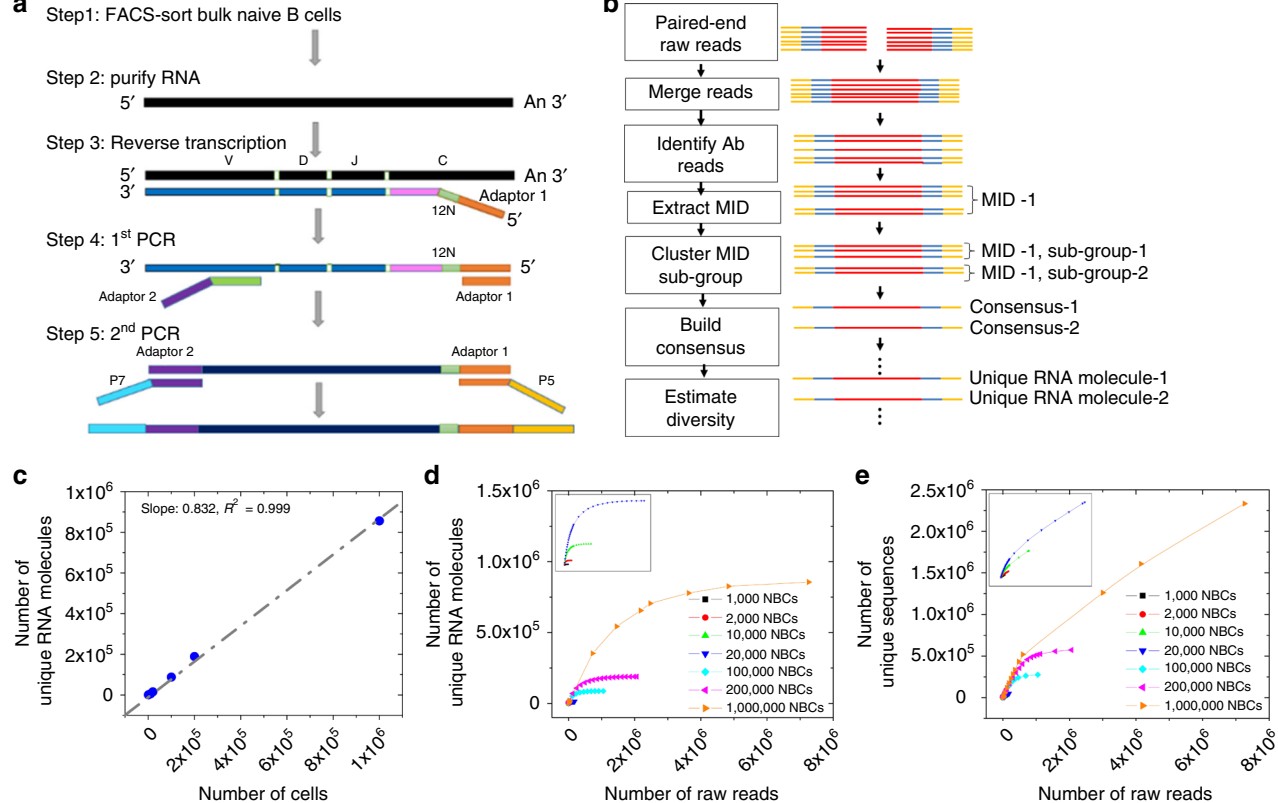

**Fig. 1** Ultra-accurate high coverage of antibody repertoire with a large dynamic range of input cells for MIDCIRS. **a** Schematic overview of tagging single immunoglobulin transcripts with MIDs. **b** Schematic overview of the informatics pipeline of MIDCIRS which includes merging paired-end reads, performing clustering to generate MID sub-groups, and building consensus sequences. **c** Correlation between number of cells and number of unique RNA molecules after using MIDCIRS. RNA from as few as 1000 to as many as 1,000,000 naive B cells was used as input material in generating the amplicon libraries. *Slope* indicates the estimated diversity coverage. **d**, **e** Rarefaction analysis of optimum sequencing depth for each sample with (**d**) and without (**e**) using MIDCIRS. NBCs naive B cells

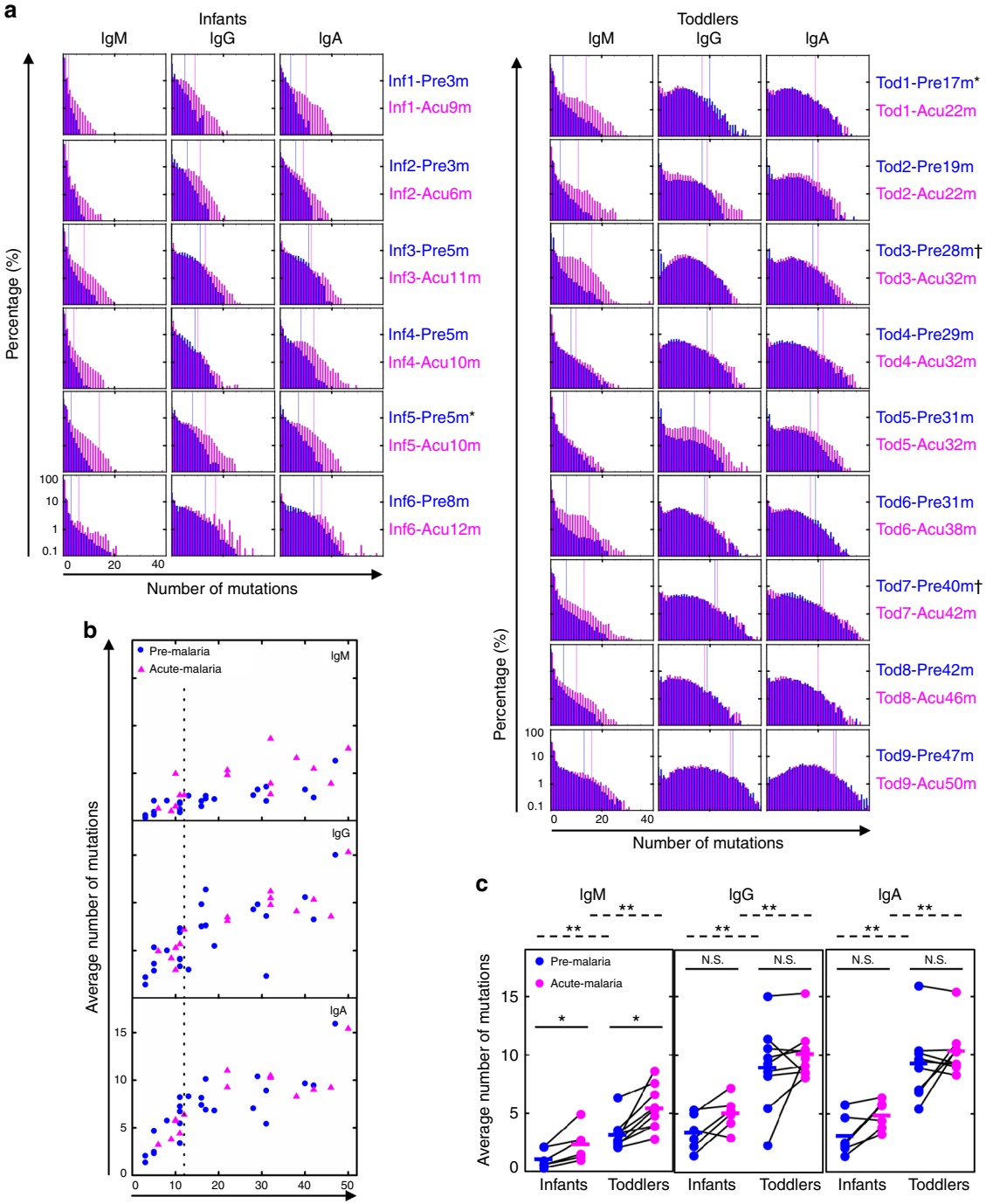

**Fig. 2** Infants and toddlers are separated into two stages based on SHM load. **a** Distribution of SHM number for infants ($N = 6$) and toddlers ($N = 9$), from whom we had paired pre-malaria (*blue*) and acute (*pink*) malaria samples, weighted by unique RNA molecules. *Blue* and *pink long vertical lines* represent the number of mutations above which 10% of sequences fall for the respective samples. * and $^{\dagger}$ demarcate samples derived from the same individuals followed for two malaria seasons. **b** Age-related average number of mutations in pre-malaria (*blue circle*, $N = 24$, $N_{Infant} = 11$, $N_{Toddler} = 13$) and acute malaria (*pink triangle*, $N = 15$, $N_{Infant} = 6$, $N_{Toddler} = 9$) samples, weighted by RNA molecules. *Dashed line* indicates the age boundary for infants (< 12 months old) and toddlers (12–47 months old). **c** Comparison of average number of mutations for paired infants and toddlers. Pre- (*blue*) and acute (*pink*) malaria samples separated by isotype; *lines* connect paired samples ($N_{Infant,paired} = 6$, $N_{Toddler,paired} = 9$). *Bars* indicate means. *$P < 0.05$, **$P < 0.01$, N.S. indicates no significant difference by two-tailed Mann–Whitney $U$ test (between age groups, *dashed lines*) or two-tailed Wilcoxon signed-rank test (between paired time points, *solid lines*). Differences in variance were not significant by squared ranks test

toddlers, the number of SHMs in reads that mutated in infants is unexpectedly high. Infants have a similar, if not higher, degree of antigen selection strength, assessed by the likelihood of amino acid-changing SHMs, compared with toddlers. Remarkably, during acute malaria, antibody lineages expand in both infants and toddlers, and this expansion is coupled with extensive diversification to the same degree as in young adults in response to acute malaria[18, 19]. Furthermore, informatically reconstructing antibody clonal lineages using sequences from both pre-malaria and acute malaria samples from the same individuals shows that

infants are capable of introducing SHMs upon a natural infection. This two time point-shared lineage analysis reveals that memory B cells isolated from pre-malaria samples in malaria-experienced individuals continue to induce SHMs upon acute malaria rechallenge and most IgM memory B cells maintain IgM, whereas a small fraction switch isotypes. In summary, using an accurate and high-coverage IR-Seq method, we discover features of the antibody repertoire that were previously unknown in infants and toddlers, shedding light on the development of the immune system and its interactions with pathogens.

## Results

**Overview of the MIDCIRS method**. MIDs have been used to track individual RNA molecules through PCR and sequencing in IR-seq to reduce error rate[10–13]. They can be designed with sufficient length, and thus diversity, to tag each individual molecule uniquely. However, this requires knowledge of the total number of RNA molecules in the sample, which requires a significant amount of time and effort to measure for each sample before library preparation. Longer MIDs are likely to decrease the reverse transcription efficiency[20, 21]. Therefore, we fixed the MID length at 12 random nucleotides and developed a generalized approach to identify each individual transcript using a sequence similarity-based clustering method to separate a group of sequencing reads with the same MID into sub-groups. Consensus sequences are then built by taking the average nucleotide at each position within a sub-group, weighted by the quality score. Each consensus sequence represents an RNA molecule, and identical consensus sequences are merged into unique consensus sequences, or unique RNA molecules (Fig. 1a, b and Methods).

**MIDCIRS yields high accuracy and coverage down to 1000 cells**. We used sorted naive B cells with varying numbers ($10^3$–$10^6$) to test the dynamic range of MIDCIRS. The resulting diversity estimates, or different types of antibody sequences, display a strong correlation with cell numbers at 83% coverage (Fig. 1c, *slope*). Previous studies have shown that about 80% of naive B cells express distinct heavy chain genes[22, 23], thus our method achieves a comprehensive diversity coverage that is much higher than other MID-based antibody repertoire-sequencing techniques[10–13].

We perform rarefaction analysis by subsampling sequencing reads to different amounts and then computing the diversity to test the effect of sequencing depth and error rate on MIDCIRS. On average, the rarefaction curves reach a plateau at a sequencing depth of around three times the cell number using MIDCIRS, suggesting that sequencing more will not discover further diversity (Fig. 1d). In contrast, without using MIDCIRS, the number of unique sequences continues to increase well beyond the number of cells for all samples (Fig. 1e). Optimum sequencing depth is likely to change depending on sample composition (e.g., PBMCs after immunization). Consistent with previous MID-based IR-seq experiments[10], MIDCIRS reduces the error rate to 1/130th of the Illumina error rate[24], providing the accuracy necessary to distinguish genuine SHMs (1 in 1000 nucleotides[25]) from PCR and sequencing errors (1 in 200 nucleotides[24]) (Supplementary Fig. 2).

**Infants and toddlers have similar VDJ usage and CDR3 lengths**. Equipped with this ultra-accurate and high-coverage antibody repertoire-sequencing tool, we applied it to study the antibody repertoire of infants and toddlers residing in a malaria endemic region of Mali. From an ongoing malaria cohort study[26], we obtained paired PBMC samples collected before and during acute febrile malaria from 13 children aged 3–47 months old

(Supplementary Fig. 3 and Supplementary Table 3). Two of the children were followed for an additional year, giving 15 total paired PBMC samples. An average of 3.8 million PBMCs per sample was directly lysed for RNA purification. All PBMCs were subjected to MIDCIRS analysis. An average of 3.75 million sequencing reads was obtained for each PBMC sample (Supplementary Table 4).

For all PBMC samples, sequencing approximately the same number of reads as the cell numbers saturates the rarefaction curve (Supplementary Fig. 4). IgM accounts for more than 50% of the repertoire in infants but reduces to less than 50% in toddlers (Supplementary Fig. 5). VDJ gene usage is highly correlated for IgM between infants and toddlers regardless of weighting the correlation coefficient by the number of sequencing reads or clonal lineages (Supplementary Fig. 6), demonstrating that the same mechanism of VDJ recombination is used to generate the primary antibody repertoire in infants and toddlers. Weighting on the number of clonal lineages in each VDJ class increases the correlation for IgG and IgA compared with weighting on the number of reads in each VDJ class (Supplementary Fig. 6). The *diagonal lines* in each *panel* indicate same sample self-correlation, and the two shorter off-diagonal lines indicate correlations from two time points of the same individual. These data recapitulate previous observations from our study in zebrafish that clonal expansion-induced differences on the number of reads in each VDJ class can confound the highly similar VDJ usage during B cell ontology[5]. In addition, infants and toddlers have similar CDR3 length distributions across the three isotypes and both time points (Supplementary Fig. 7), consistent with recent studies of PBMCs from 9-month-olds infants[14, 15] and adults[27, 28], and confirming the previous results that an adult-like distribution of CDR3 length is achieved around 2 months of age[29].

**Both infants and toddlers have unexpectedly high SHM**. SHM is an important characteristic of antibody repertoire secondary diversification due to antigen stimulation[30]. Although it has been demonstrated before that infants have fewer mutations in their antibody sequences than toddlers and adults, the limited number of sequences for only a few V genes does not provide convincing evidence of the levels of SHM in infants[31]. A recent study using the first generation of IR-seq showed that two 9-month-old infants averaged at least six SHMs in IgM of an average length of 500 nucleotides[14]. These numbers are equivalent to, if not higher than, reported SHM rates in IgM sequences from healthy adults day 7 post influenza vaccination[10] and are much higher than a low-throughput infant study using a few V genes and limited antibody sequences[32]. Owing to the inherent errors associated with the first generation of IR-seq as discussed above, it is possible that PCR and sequencing errors had a role[14]. In addition, it remains unclear whether infants (< 12 months old) are able to generate a significant number of mutations in response to infection, which would demonstrate their capacity to diversify the antibody repertoire[33].

Here, we show that infants (< 12 months old) and toddlers (12–47 months old) reach an unexpectedly high level of SHMs in all three major isotypes, particularly IgG and IgA[34] (Fig. 2a). Although the mutation distributions remain in the low end of the spectrum for IgM, the number of mutations is significantly higher in IgG and IgA for both age groups. The threshold for the 10% most highly mutated unique RNA molecules is around 10 in infant IgG and IgA sequences (Fig. 2a, infants, *right* of the *blue long vertical lines*) and around 20 in toddler IgG and IgA sequences (Fig. 2a, toddlers, *right* of the *blue long vertical lines*). To minimize any possible inflation of SHMs, we excluded all

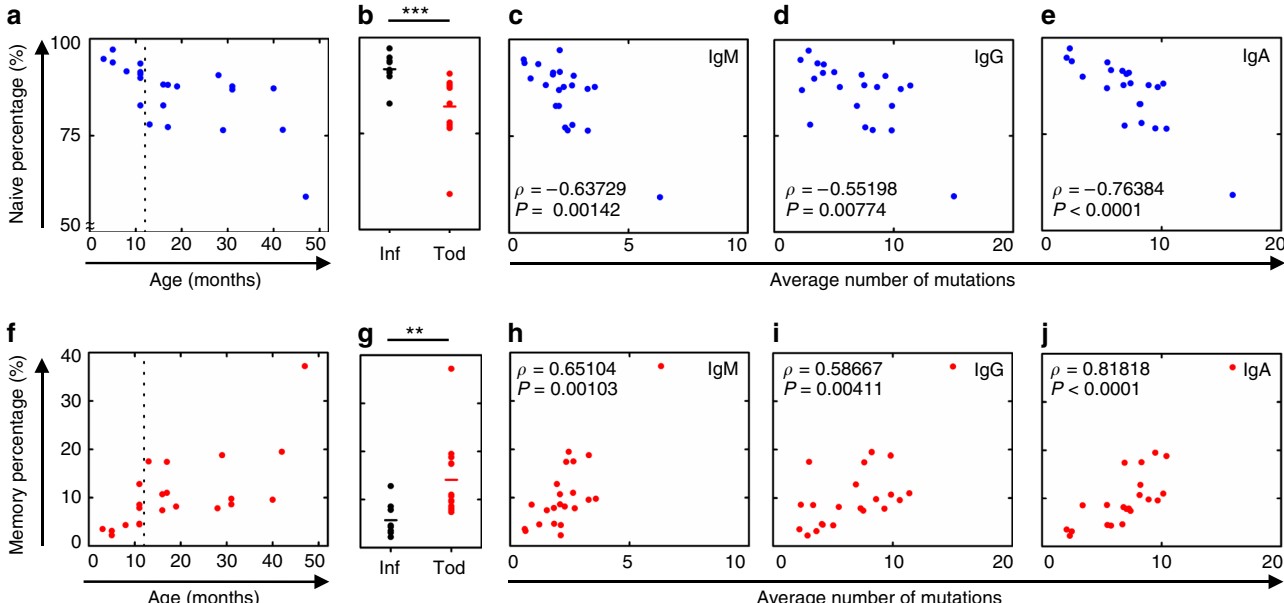

**Fig. 3** Decrease of naive B cell and increase of memory B cell percentages show a two-stage trend and correlate with SHM load. **a** Naive B cell percentages of total B cells from the pre-malaria samples (N = 22) vary with age. *Dashed vertical line* depicts the cutoff between infants and toddlers. **b** Naive B cell percentages of total B cells compared between infants (*black*, N = 9) and toddlers (*red*, N = 13). **c–e** Naive B cell percentages correlate with average number of mutations (SHM load) in IgM (**c**), IgG (**d**), and IgA (**e**) sequences from bulk PBMCs in pre-malaria samples (N = 22). **f** Memory B cell percentages of total B cells from the pre-malaria samples (N = 22) vary with age. *Dashed vertical line* depicts the cutoff between infants and toddlers. **g** Memory B cell percentages of total B cells compared between infants (*black*, N = 9) and toddlers (*red*, N = 13). **h–j** Memory B cell percentages correlate with average number of mutations (SHM load) in IgM (**h**), IgG (**i**), and IgA (**j**) sequences from bulk PBMCs in pre-malaria samples (N = 22). **b**, **g** *Bars* indicate means; **P < 0.01, ***P < 0.001, two-tailed Mann–Whitney U test. **c–e**, **h–j** ρ and P values determined by Spearman's rank correlation listed in each panel

sequences that were mapped to novel alleles, which were identified by both TIgGER[35] and inspecting IgM sequences (Methods). These putative novel alleles account for 8% of all unique sequences on average (Supplementary Table 5). Naive B cells from these same patients, sorted as a control, harbor only 0.55 mutations on average, as expected (Supplementary Table 6). Upon acute malaria infection, the SHM histogram shifts rightward for almost all isotypes in almost all individuals (Fig. 2a, the *right shift* of *pink long vertical line* compared to *blue long vertical line*), including infants. These results demonstrate high levels of SHM that exceed what have been documented previously[32, 34].

**SHM load is distinct between infants and toddlers**. The differences in the shapes of SHM distributions of infants and toddlers, steadily decreasing from unmutated for infants in all three isotypes while peaking around 10 for toddlers in IgG and IgA (Fig. 2a), suggest that the total SHM load might reflect the history of interactions between the antibody repertoire and the environment, including malaria exposure. As the malaria season is synchronized with the 6-month rainy season (Supplementary Fig. 3), and > 90% of the individuals in this cohort are infected with *P. falciparum* during the annual malaria season[26], we hypothesized that the SHM load would increase with age. However, we found that the SHM load rapidly increases with age in infancy and then appears to plateau around 12 months of age in an initial smaller set of children with paired pre-malaria and acute malaria PBMC samples (Supplementary Fig. 8). We then added nine pre-malaria samples around the infant and toddler transition (5 of 11 months old and 4 of 13 to 17 months old). The two-staged trend of SHM load remains for all three isotypes (Fig. 2b), with samples around the transition having the largest variation. Detailed comparisons show that, consistent with the two-stage trend, toddlers have a higher SHM load compared with infants for all three isotypes at both pre-malaria and acute malaria

time points (Fig. 2c, comparison between age groups). Although there is a significant increase on SHM load upon acute malaria infection in IgM for both infants and toddler, bulk PBMC analysis does not show a significant increase in IgG or IgA, possibly because of the already elevated SHM base level. This, along with the two-stage trend (Fig. 2b), suggests that 12 months is an important developmental threshold for secondary antibody repertoire diversification: before this threshold, the global repertoire is quite naive but can quickly diversify upon a natural infection.

**Higher memory B cell percentage results in higher SHM load**. This unexpected developmental threshold of secondary antibody repertoire diversification prompted us to focus on B cell subset composition changes and ask whether they correlate with this two-staged SHM load. Flow cytometry analysis reveals that naive B cells decrease from about 95% in 3-month-old infants to about 80% in toddlers (Fig. 3a). Conversely, memory B cells increase from about 4% in 3-month-old infants to about 15% in toddlers (Fig. 3f). As the two-stage SHM load analysis suggests, 12 months appears to divide the samples into two age groups, with a large variation at the infant to toddler transition and in the toddler group. Infants have a significantly more naive B cells and fewer memory B cells than toddlers (Fig. 3b, g). Plasmablast percentages fluctuated in a much smaller range (Supplementary Fig. 10). With a similar two-staged trend observed for B cell subset percentages, we hypothesized that the B cell subset percentage would correlate with SHM load. Indeed, further analysis shows that the decrease in naive B cell percentage and the increase in memory B cell percentage correlate well with SHM load across IgM, IgG, and IgA isotypes (Fig. 3c–e, h–j), which supports our initial hypothesis that 12 months separates infants from toddlers in both SHM load and B cell composition changes. These data suggest that memory B cells contribute significantly to the developing

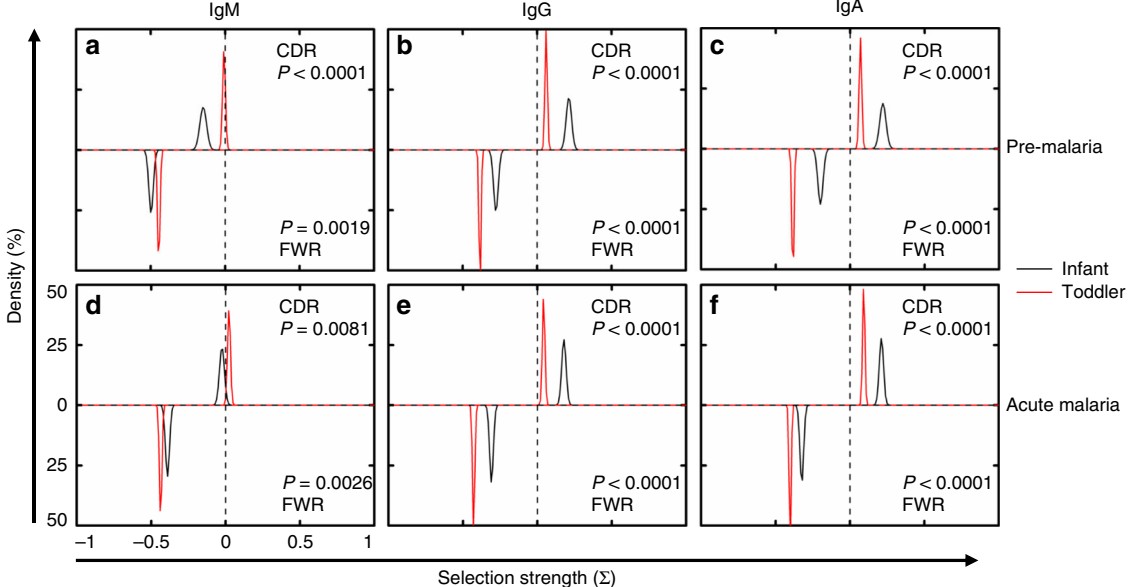

**Fig. 4** Antigen selection strength comparisons between infants and toddlers. Selection strength distributions, as determined by BASELINe[37], were compared between infants (*black*) and toddlers (*red*) for PBMCs from pre-malaria (**a–c**) ($N_{infant} = 6$, $N_{toddler} = 9$) and acute malaria (**d–f**) ($N_{infant} = 6$, $N_{toddler} = 9$) time points, separated by isotype: **a**, **d** IgM; **b**, **e** IgG; and **c**, **f** IgA. Selection strength on CDR (CDR1 and 2, *top half* of each *panel*) and FWR (FWR2 and 3, *bottom half* of each *panel*) for unique RNA molecules was calculated. CDR3 and FWR4 were omitted due to the difficulty in determining the germline sequence. FWR1 for all sequences was also omitted because it was not covered entirely by some of the primers. *P* value calculated as previously described[37]

antibody repertoire, and their composition is essential in secondary antibody repertoire diversification.

**SHMs are similarly selected in infants and toddlers**. One of the key features of antibody affinity maturation is antigen selection pressure imposed on an antibody, which is reflected in the enrichment of replacement mutations[36] in the CDRs, the parts of the antibody that interact with antigens, and the depletion of replacement mutations in the framework regions (FWRs), the parts of the antibody responsible for proper folding. The unexpectedly high level of SHMs observed in infants prompted us to ask whether those SHMs have characteristics of antigen selection, as seen in older children and adults. As previous studies have shown that infants have limited CD4 T cell responses and neonatal mice exhibit poor germinal center formation[17], we hypothesized that infant antibody sequences would display weaker signs of antigen selection. Here, we use a recently published tool, BASELINe[37], to compare the selection strength. BASELINe quantifies the likelihood that the observed frequency of replacement mutations differs from the expected frequency under no selection; a higher frequency implies positive selection and a lower frequency implies negative selection, and the degree of divergence from no selection relates to the selection strength. Surprisingly, despite infants harboring fewer overall mutations, these mutations are positively selected in the CDRs and negatively selected in the FWRs in both IgG and IgA (Fig. 4b, c, e, f). Contrary to the hypothesis that infants would have a lower selection strength than toddlers, for both IgG and IgA, infants actually have a higher selection strength at both pre-malaria and acute malaria time points (Fig. 4). The lower selection strength in infant IgM sequences at the pre-malaria time point is significantly higher during acute malaria infection (Fig. 4a, d, CDR *black curves* between two time points, *P* < 0.0001 (numerical integration, as previously described[37])), suggesting that the significant increase in SHM is antigen-driven and selected upon. To compare with a large amount of historical adult data, we

calculated replacement to silent mutation ratios (*R/S* ratios), which are about 2-3:1 in FWRs and 5:1 in CDRs for both infants and toddlers (Supplementary Table 7). These results are similar to adults[36, 38–40], and much higher than what has been reported for children previously using a very limited number of sequences[41]. We also noticed that *R/S* ratio in the FWRs of IgM was much higher in infants, contrary to the BASELINe results, which highlights the importance of incorporating the expected replacement frequency when considering selection pressure. These results suggest that as an end result of interactions between antigen selection and SHM, the degree of antibody amino acid changes is comparable in infants, toddlers, and adults. It also suggests that cellular and molecular machineries for antigen selection are already in place in infants.

**Clonal lineages diversify upon acute febrile malaria**. The exhaustive sequencing data obtained by MIDCIRS offers the possibility to reconstruct clonal lineages that trace B cell development. Clonal lineages contain different species of unique antibody sequences that could be progenies derived from the same ancestral B cell. B cell clonal lineage analysis has been used to track affinity maturation and sequence evolution of HIV broadly neutralizing antibodies[42, 43]. Using a clustering method with a pre-determined threshold (90% similarity on nucleotide sequence at CDR3), we previously demonstrated that B cell clonal lineages could be informatically defined and contain pathogen-specific antibody sequences[6]. In addition, the clonal lineage analysis also highlighted the lack of antibody diversification in the elderly after influenza vaccination[6]. Using the same approach and a similar threshold[6, 44], we aimed to answer whether infants and toddlers are able to diversify antibody clonal lineages in response to infection and, if so, whether they have a similar ability to do so, which was previously impossible to answer due to technical limitations. To do this, we first visualized structures of informatically defined clonal lineages for the entire antibody repertoire (Supplementary Fig. 11). Each oval lineage

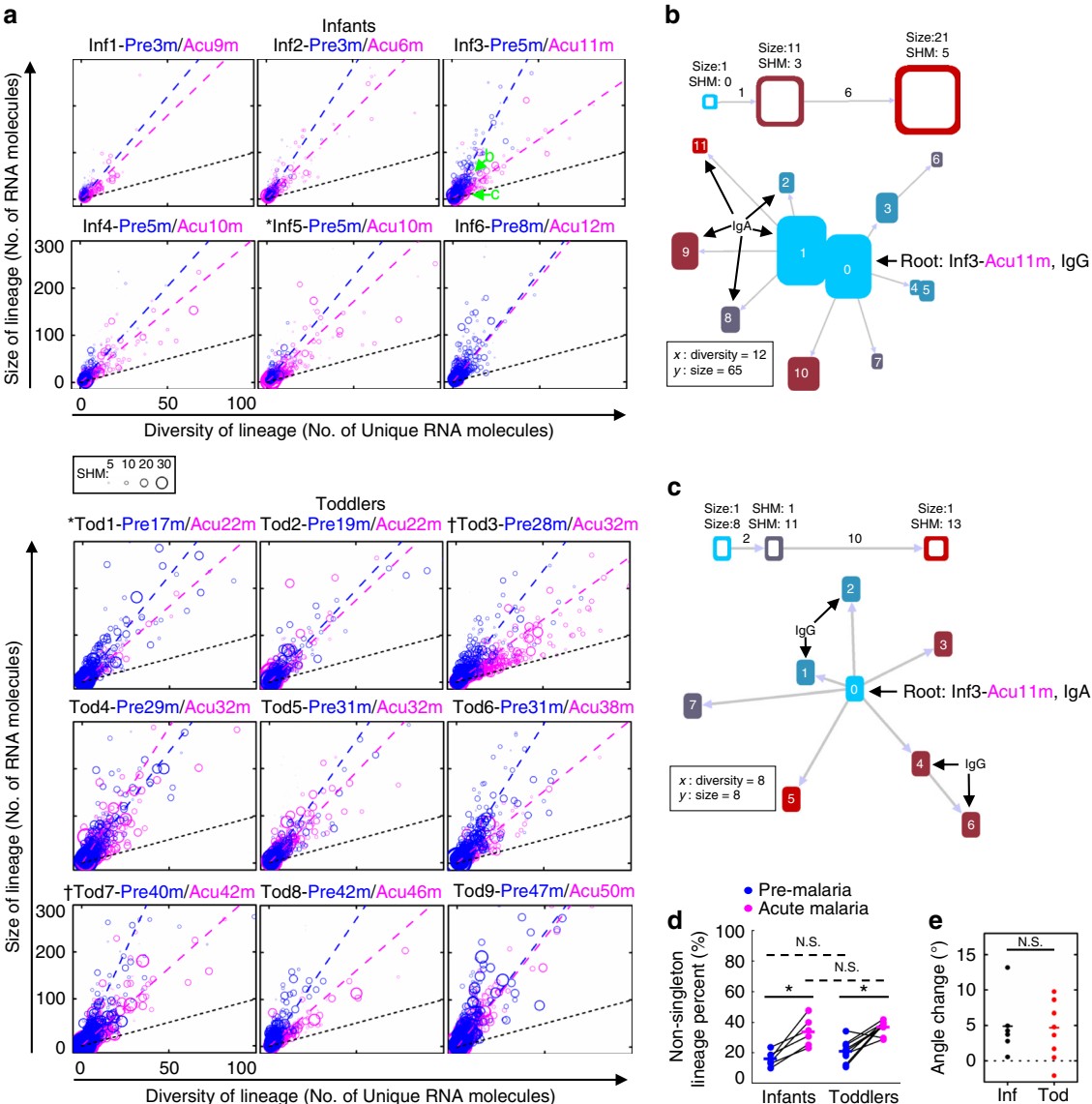

**Fig. 5** B cell lineage complexity change under malaria stimulation. **a** Diversity and size of B cell lineages for infants ($N = 6$) and toddlers ($N = 9$) from whom paired PBMC samples at pre-malaria (*blue*) and acute malaria (*pink*) were obtained. *Each circle* represents an individual lineage. The area of each *circle* is proportional to the SHM load. Labeled *green arrows* indicate representative lineages whose intra-lineage structures were shown in detail in (**b**) and (**c**). Each circle's $x$ and $y$ coordinates were determined by its diversity (the number of unique RNA molecules in a lineage) and size (the number of total RNA molecules in a lineage), respectively. *Blue* and *pink dashed lines* represent the linear fit for pre-malaria and acute malaria lineages, respectively. *Black dashed lines* indicate $y = x$ parity, such that lineages lying on the parity line are comprised entirely of unique RNA molecules with minimum clonal expansion, such as lineage in (**c**). On the other hand, lineages comprised of clonally expanded RNA molecules are close to the $y$ axis, such as lineage (**b**). **b**, **c** Each node is a unique RNA molecule species. The *height* of the node corresponds to the number of RNA molecules of the same species, the *color* corresponds to number of nucleotide mutations, and the distance between nodes is proportional to the Levenshtein distance between the node sequences, as indicated in the legend above each lineage. All unlabeled nodes share the isotype with the root. **d** The non-singleton lineage percent (lineages comprises at least two RNA molecules) between infants and toddlers at pre-malaria (*blue*) and acute (*pink*) malaria. *$P < 0.05$ by two-tailed Wilcoxon signed-rank test (between time points, *solid lines*); N.S. indicates no significant difference by two-tailed Mann–Whitney $U$ test (between age groups, *dashed lines*). **e** The difference of linear regression slopes (*angles*), or degree of diversity change, between pre-malaria and acute malaria for infants (*black*) and toddlers (*red*). N.S. indicates no significant difference by two-tailed Mann–Whitney $U$ test. *Bars* indicate means. Differences in variance were not significant by squared ranks test

map represents an individual PBMC sample at one time point. Densely packed individual lineages are not easily identified visually in Supplementary Fig. 11; however, *dark areas* indicate that clonal lineages are already complex in this cohort of infants as young as 3 months old and can be further diversified upon acute febrile malaria.

The densely packed lineages could result from large lineage sizes (one unique RNA molecule with many copies), large lineage diversities (many unique RNA molecules), or a combination of

the two. To closely examine the possible differences in the degree of this intra-clonal lineage expansion and diversification between infants and toddlers, especially upon acute febrile malaria, we projected the global lineage structure (Supplementary Fig. 11) onto diversity and size of lineage axes (Fig. 5a). *Each circle* represents an individual lineage, with the area of the circle proportional to the SHM load (average mutations of the lineage). This analysis effectively captures five parameters that quantify lineage complexity in a sample: number of total clonal lineages

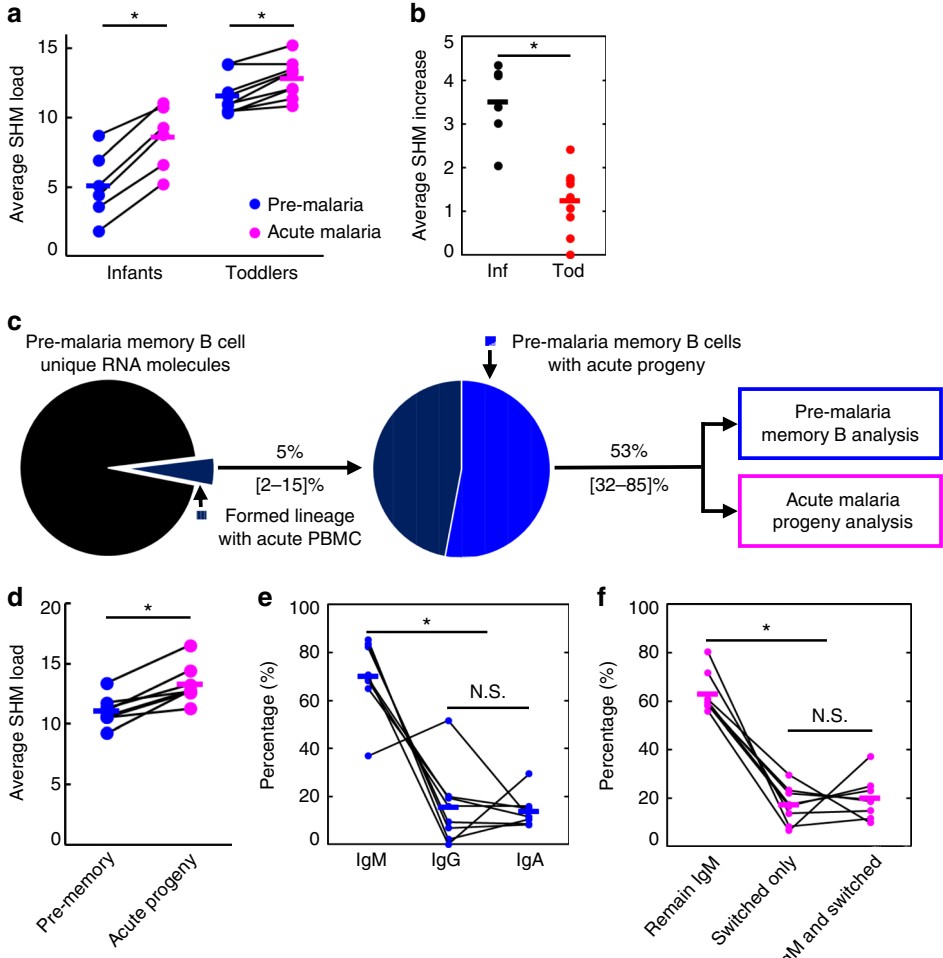

**Fig. 6** Two time point-shared lineage analysis reveals SHM increment during acute malaria infection. **a** Average SHM for sequences from pre-malaria (*blue*) and acute malaria (*pink*) time points within lineages containing sequences from both time points for infants (*N* = 6) and toddlers (*N* = 9). **b** Average SHM increase upon acute malaria infection for infants (*black*) and toddlers (*red*) from (**a**). **c** Flow diagram for two time point-shared lineage containing pre-malaria memory B cell identification and acute progeny analysis. Percentages represent the average percent of unique sequences classified by the indicated slice, range in *brackets*. **d** Average SHM load for pre-malaria memory B cells with acute progeny (*blue*) and their acute progenies (*pink*) for malaria-experienced toddlers with FACS-sorted pre-malaria memory B cells (*N* = 8). **e** Isotype distribution of pre-malaria memory B cells with acute progeny. **f** Isotype fate of acute progenies stemming from IgM pre-malaria memory B cells. *Lines* connect the same individuals. *Bars* indicate means. **a**, **d**–**f** *P* < 0.05, N.S. indicates not significant by two-tailed Wilcoxon signed-rank test. **b** *P* < 0.05 by two-tailed Mann–Whitney *U* test

(number of circles), diversity of each lineage (*x*-axis position, number of unique RNA molecules in a lineage), size of each lineage (*y*-axis position, number of total RNA molecules in a lineage), SHM load of each lineage (area of circle, key is located in between the infant and toddler panels in Fig. 5a), and the extent of clonal expansion of each lineage (distance from *y* = *x* parity line; no clonally expanded RNA molecules within a lineage if it is on parity line or pure clonal expanded RNA molecules if it is in the *top left quadrant* of each panel).

Figure 5b, c are two example lineages selected to display the full lineage structures to demonstrate a lineage with diversification and clonal expansion (Fig. 5b refers to *green letter* "b" indicated in Fig. 5a, Inf3) and another one with diversification but without clonal expansion (Fig. 5c refers to *green letter* "c" indicated in Fig. 5a, Inf3). Both are represented by a single *circle* in Fig. 5a, but their locations in Fig. 5a depend on the numbers of RNA molecules (*y* axis) and numbers of unique RNA molecules (*x* axis). Lineage "c" (c in Fig. 5a, Inf3, zoomed in view in Fig. 5c) that lies away from the origin and near the *black y* = *x* parity line consists of 8 unique sequences, each represented by only one RNA molecule, indicating extensive lineage diversification but no

clonal expansion. Lineage "b" (b in Fig. 5a, Inf3, zoomed in view in Fig. 5b) that lies far from the parity line is dominated by two unique RNA molecules each with about 20 copies (Fig. 5b, *height* of nodes), indicating extensive clonal expansion of particular sequences in addition to diversification. Changing lineage forming threshold from 90 to 95% does not change the overall structure of the lineages (Supplementary Fig. 12).

This five-dimension lineage analysis reveals that infants as young as 3 months old can generate extensive lineage structures, with many lineages containing more than 20 different types of antibody sequences and 50 RNA molecules (Fig. 5a). Toddlers have many more lineages with higher levels of both size and diversity. However, in both infants and toddlers, the majority of clonal lineages are singleton lineages consisting of only one RNA molecule (Fig. 5d), consistent with the flow cytometry analysis that the bulk of the B cell repertoire is naive in these young children (Fig. 3). Upon acute malaria infection, the fraction of non-singleton lineages increases in both infants and toddlers (Fig. 5d).

To tease out whether these non-singleton lineages diversify or clonally expand upon acute infection, we fit linear regressions to

the lineage diversity-size plots. An immune response against an infection can have a twofold effect on the lineage landscape: antigen stimulation can cause clonal expansion, which would shift the lineage up on the $y$ axis, and SHM and affinity maturation, which would shift the lineage to the right on the $x$ axis. This balance between clonal expansion and diversification is depicted by the slope of the linear regression (Fig. 5a, *dashed blue lines* for pre-malaria samples and *dashed pink lines* for acute malaria samples). We hypothesized that the lower absolute SHM load of infants would imply a defect in the ability to diversify clonal lineages in response to infection, leading the slope change from pre-malaria to acute malaria to be low (a small angle between *blue* and *pink dashed lines*) or even negative (*pink dashed line* is closer to $y$ axis than *blue dashed line*). Surprisingly, our analysis shows that infants diversify their clonal lineages in a similar manner as toddlers in response to acute malaria (Fig. 5e). As singleton lineages do not bear any weight on the linear regression, our analysis shows that the increasing fraction of non-singleton lineages upon malaria infection is similarly diversified between infants and toddlers, which is also similar to a young adult at pre-malaria and acute malaria (Supplementary Fig. 14). However, this sharply contrasts with what we had previously observed in the elderly following influenza vaccination, where clonal expansion dominated[6]. Among clonally expanding and diversifying B cell clones during an infection, only a subset of the cells comprising the clonal burst remain once the infection has been cleared. Thus, the characteristic change in the lineage size/diversity linear regression slope upon infection is expected to subside as time passes since the acute infection. Indeed, comparing the pre-malaria lineage size/diversity linear regression slopes reveals no difference between infants (who have not experienced malaria before) and toddlers (who have experienced malarias in previous years) (Supplementary Fig. 13). These results highlight the unexpected capability of young children's antibody repertoire in response to a natural infection.

**SHM load increases upon an acute febrile malaria infection.** The plateau we observed on SHM load in toddlers at both pre-malaria and acute malaria (Fig. 2b) and the lack of a SHM difference in IgG and IgA between pre-malaria and acute malaria (Fig. 2c) seems to suggest that the experienced part of the repertoire does not respond to malaria infection by inducing SHM. However, it could be that only a portion of the bulk antibody repertoire responds to the infection and there is already a high level of baseline SHMs as revealed by the histogram analysis (Fig. 2a). As we saw the lineage diversification upon malaria infection in Fig. 5, we hypothesized that examining the SHMs from sequences in two time point-shared lineages (lineages containing both pre-malaria and acute malaria sequences) would enable us to quantify the infection-induced SHM increase from the highly mutated background. To test this, we pooled all sequences from both time points, including sorted memory B cells at pre-malaria, and generated lineages again using the 90% similarity threshold at CDR3[6, 44]. We are able to find two time point-shared lineages in all individuals analyzed (Supplementary Table 8). Consistent with the observation that toddlers already have a diverse and expanded antibody repertoire compared to infants, there are more shared lineages in toddlers than infants (Supplementary Table 8). We tallied SHMs for sequences from pre-malaria and acute malaria in the two time point-shared lineages separately. Consistent with our hypothesis, both infants and toddlers significantly increase SHM upon infection (Fig. 6a). Indeed, toddlers had a higher pre-malaria SHM level compared to infants (Fig. 6a, *blue symbols*). To our surprise, infants were able to induce more SHMs compared to toddlers (Fig. 6b). These data

suggested that indeed both infants and toddlers induce SHMs upon malaria infection.

**Memory B cells further diversify upon malaria rechallenge.** The importance of IgM-expressing memory B cells has been reported in mice in several recent studies[45–48], including a mouse model of malaria infection[49]. However, fewer studies have examined these cells in humans[50], and their composition and role in repertoire diversification upon rechallenge remains elusive. It is widely believed that they may retain the capacity to introduce further mutations and class switch[45, 46, 49, 51]. However, sequence-based clonal lineage evidence is lacking. The paired samples before and during acute malaria from toddlers who experienced malaria in previous years provided an opportunity to investigate the role of memory B cells in repertoire diversification upon rechallenge in children.

Here, we focus on two time point-shared lineages that harbor sequences from pre-malaria memory B cells. Given the significant increase of SHM we identified at acute malaria sequences over pre-malaria sequences in two time point-shared lineages (Fig. 6a), we reasoned that the high repertoire coverage of MIDCIRS should enable us to identify a large number of two time point-shared lineages that contain these memory B cells, and these memory B cells should have mutated progenies at the acute malaria time point. To ensure that we identify sequence progenies of these pre-malaria memory B cells, we employed an antibody lineage structure construction algorithm, COLT, that we recently developed[52]. COLT considers isotype, sampling time, and SHM pattern when constructing an antibody lineage, which allows us to trace, at the sequence level, the acute progeny of these memory B cells. As illustrated by Supplementary Fig. 15, this COLT-generated lineage tree depicts a pre-malaria memory B cell sequence serving as a parent node to sequences derived from the acute malaria time point. This analysis is much more stringent in identifying sequence progenies than simply judging if a pre-malaria memory B cell sequence is grouped with acute malaria PBMC sequences.

On average, 5% of unique sequences from 10,000 sorted memory B cells form lineages with acute malaria PBMC sequences (Fig. 6c, *dark blue slice* of the first pie). COLT[52] analysis on these pre-malaria memory B cell-containing lineages shows that 53% contain traceable progeny sequences from the acute malaria PBMCs (Fig. 6c, *lighter blue slice* of the second pie). Overall, there is a significant increase of SHM in these acute malaria progenies compared with their ancestor pre-malaria memory B cells (Fig. 6d). Consistent with previous studies[45, 46, 51], these progeny-bearing pre-malaria memory B cells express all three major isotypes, with IgM being the dominant species (Fig. 6e). Investigating their isotype switching capacity reveals that about 60% of the IgM pre-malaria memory B cells maintain IgM as progenies; however, about 20% only have isotype-switched progenies detected, whereas the remaining 20% have both IgM and isotype-switched progenies (Fig. 6f). These pre-malaria IgM memory B cells largely retain IgM expression while further introducing SHM upon rechallenge. Thus, these analyses show multi-facet diversification potential of young children's memory B cells in a natural infection rechallenge.

## Discussion

About 13,000 children under 1 year old die every day world-wide[53], and most of these deaths are caused by infection[17]. It has long been recognized that children's immune systems are immature at birth and require time to develop to provide protection against pathogens or respond to vaccines. However, few studies have focused on children's antibody repertoire

development, diversification, and response to infection. Knowledge in this area holds great interest to vaccine development and vaccination strategy design. This is especially urgent for malaria, as it still kills about half a million children each year[54], and the most advanced malaria vaccine confers only partial, short-lived protection in African children[55].

However, studying the antibody repertoire in young children is challenging in several regards: first, lack of analytical tools to exhaustively study the antibody repertoire from small volumes of blood; second, lack of informatic analysis tools to turn high-throughput data into knowledge; and third, the rarity of a set of samples from young children obtained before and at the time of a natural infection. To address these challenges, we developed a highly accurate and high-coverage immune repertoire-sequencing tool, MIDCIRS, to analyze antibody repertoire development, diversification, and capacity to respond to a natural infection in children who were experiencing acute febrile malaria.

Previous studies showed that there was evidence of SHM and antigen selection in infants 8 months of age or older by examining a few V-gene alleles[31]. However, it is not clear how widespread SHMs are in infant antibody repertoires and to what degree SHMs can be introduced in response to an infection. By using a comprehensive and unbiased analysis, here we show that infants as young as 3 months old can have 10% of sequences with five or more mutations, and they can further introduce mutations upon an acute febrile malaria infection to well over 20 SHM per 270 nt heavy chain V region. Compared with toddlers, there is a separation on SHM load around 12 months: this number gradually increases before 12 months and stays at a plateau after that regardless of repeated malaria incidents. Consistent with this trend is the similar pattern observed in the increase in the percentage of memory B cells and corresponding decrease in percentage of naive B cells with age: both plateaued after 12 months of age. Accordingly, SHM load in IgM, IgG, and IgA correlates with the percentages of naive and memory B cells. Surprisingly, regardless of the lower mutation load in infants, their mutations are similarly, if not more strongly, selected as those of toddlers, suggesting that the molecular machineries and other cellular components involved in antibody selection are already developed in infants. In future analyses, it will be of interest to tease out the mechanistic contributions to a two-stage increase of average mutation number, in particular, the role of T cell help and germinal center formation. Regardless of these detailed mechanisms, it is clear infants can perform antibody selection as well as toddlers and adults, which provides some assurance of the effectiveness of vaccination in young children.

Another interesting finding is that infant antibody repertoires are capable of diversification, as seen through the clonal lineage analysis. We show that both infants and toddlers diversify their repertoire to the same degree upon an acute febrile malaria infection. Although adults have a larger range on the diversity of the lineage because they have accumulated more mutations, the degree of repertoire diversification in infants and toddlers is similar to what adults experience at an acute febrile malaria. This contrasts how elderly individuals preferentially increase the size of these lineages due to clonal expansion after influenza vaccination[6]. Altogether, these data provide evidence that infant antibody repertoires are similarly capable of responding to malaria as toddlers and adults.

Analyzing antibody lineages formed by sequences from both pre-malaria and acute malaria samples, we see an increase in SHM upon malaria infection. Although the high background SHMs at pre-malaria prevented us from detecting an increase of SHM upon malaria infection in the bulk repertoire except in IgM, our two time point-shared lineage analysis shows both infants and toddlers increase SHMs upon acute malaria infection. This

highlights the power of combining informatics analysis with improved technology—the magnitude of SHM increase in toddlers is relatively small compared with infants, which would have been missed if MIDs were not used. Similar problems may prevent a recent study from identifying an increase of SHM in healthy adults who received influenza vaccination[50]. It is also possible that the perturbation of a natural infection is much stronger than vaccination. The evidence we observe that infants have a similar propensity to diversify clonal lineages, perform antigen selection, and increase SHM load in acute malaria also provides a comprehensive assessment of the capacity of infants' antibody repertoire to respond to a natural infection. These results also provide some assurance that infants are capable of responding to external stimuli and develop significantly diversified antibody lineages. These, along with the observation that administering anti-parasite drugs in the first 10 months of life correlated with higher malaria incidence in 2nd year of life[56], suggest that repeated vaccination might mimic the naturally acquired clinical malaria resistance that arises through repeated malaria exposures[26, 56].

Using progeny tracing in two time point-shared lineages containing pre-malaria memory B cells, we were able to detect total of 1799 lineages that contained acute progenies. We analyzed conventional memory B cells because the atypical memory B cell frequency is low in this cohort of young children (Supplementary Fig. 9). This is possibly due to the relatively few malaria incidents in these young children, as opposed to adults and older children that we and others have analyzed where atypical memory B cell increases prevalence as a result of many years of chronic malaria exposure[57, 58]. In depth analysis shows both continued SHM and isotype switching in memory B cells in response to a natural infection rechallenge. Although we do not know the cell phenotype of these progeny sequences, it is reasonable to speculate that some of them could have developed into plasmablasts. A previous study has shown that plasmablasts have about 30 to 35 times more transcripts per cell compared to other peripheral B cell populations[59]. Even when we relaxed the threshold and counted all unique sequences that had more than 10 copies as plasmablasts, at most 2.1% of unique sequences in these pre-malaria memory B cell-derived progenies could be inferred as plasmablasts (Supplementary Fig. 16). Applying the same threshold to bulk PBMC sequences from acute malaria time point resulted in a similar percentage of plasmablasts-derived sequences, which is consistent with about 2.2% of plasmablasts by flow cytometry analysis. Altogether, these data suggest that most of the progenies from pre-malaria memory B cells remain memory B cells and continued to increase SHMs in these young children.

In summary, we presented an improved method of utilizing MIDs to significantly enhance the coverage and dynamic range of IR-seq. Using it, we systemically studied the antibody repertoire in malaria-exposed infants and toddlers and discovered several aspects of repertoire development, diversification, and capacity to respond to an infection that were not known before, which provides not only new parameters and approaches in quantifying vaccine efficacy beyond traditional serological titer but also venues for future studies of detailed molecular and cellular mechanisms that drive antibody repertoire differences between infants and toddlers.

## Methods

**Study design and cohort**. Human PBMCs for method validation were purified from de-identified blood bank donor samples. This protocol was approved by the Institutional Review Board of the University of Texas at Austin as non-human subject research.

Infant and toddler PBMC samples from 22 residents of Kalifabougou, Mali, ranging from 3 months old to 47 months old, were collected from an ongoing malaria cohort study[26] and analyzed as summarized in Supplementary Table 3.

Enrollment exclusion criteria were hemoglobin level < 7 g/dL, axillary temperature ≥ 37.5°C, acute systemic illness, use of antimalarial or immunosuppressive medications in the past 30 days, and pregnancy. The research definition of malaria was an axillary temperature of ≥ 37.5°C, ≥ 2500 asexual parasites per µL of blood, and no other cause of fever discernible by physical exam. The Ethics Committee of the Faculty of Medicine, Pharmacy, and Dentistry at the University of Sciences, Technique, and Technology of Bamako, and the Institutional Review Board of the National Institute of Allergy and Infectious Diseases, National Institutes of Health, approved the malaria study, from which we obtained frozen PBMCs. Written informed consent was obtained from adult participants and from the parents or guardians of participating children. The study is registered in the ClinicalTrials.gov database (NCT01322581).

For this study, individuals were chosen based on the availability of frozen PBMCs in the age range specified. Blood draws were taken before the rainy season, when mosquitos are not rampant and the cases of malaria are low, and during acute febrile malaria (Supplementary Fig. 3). Samples were labeled for analysis by the group (Inf/Tod), patient ID, time point, and age at blood draw (in months), e.g., Inf1-Pre3m represents the pre-malaria time point for infant 1 who was 3 months old at the time of blood draw (Supplementary Table 3). Samples collected before the beginning of the rainy season that tested PCR negative for *Plasmodium falciparum* and *Plasmodium malariae* were designated "pre-malaria". Samples collected 7 days into acute febrile malaria infection were designated "acute malaria". Among them, two individuals were tracked for two consecutive malaria seasons and only the pre-malaria time point was obtained and analyzed for nine individuals around the infant to toddler transition, as indicated in Supplementary Table 3. Authors were not blinded to the age-group allocation or to the sample collection time.

**Cell sorting**. Frozen PBMCs were thawed, washed once in RPMI, and then processed for cell staining and sorting following standard FACS staining protocol[60, 61]. Naive B cells were FACS-sorted based on the phenotype of CD3⁻CD19⁺CD20⁺CD27⁻CD38^low. For malaria samples, up to 5,000,000 PBMCs were lysed directly. The remaining PBMCs were analyzed via flow cytometry; plasmablasts were gated based on the phenotype of CD4⁻CD8⁻CD14⁻CD56⁻CD19⁺CD27^bright CD38^bright and memory B cells were gated based on the phenotype of CD4⁻CD8⁻CD14⁻CD56⁻CD19⁺CD20⁺CD27⁺CD38^low. Naive and memory B cells were not gated on IgD or CD21, so only conventional memory B cells were analyzed. Sorted cells were lysed in RLT Plus buffer (Qiagen) supplemented with 1% β-mercaptoethanol (Sigma). The following antibody clones were obtained from Biolegend and used at 1:25 dilution: Alexa Fluor 488 OKT3 (CD3), Alexa Fluor 700 RPA-T4 (CD4), Alexa Fluor 488 HCD14 (CD14), APC-Cy7 2H7 (CD20), PE-Cy7 O323 (CD27), APC HIT2 (CD38), Alexa Fluor 488 MEM-188 (CD56), and Brilliant Violet 605 IA6-2 (IgD). The following antibody clones were obtained from BD Biosciences: PE-CF594 RPA-T8 (CD8), Brilliant Ultraviolet 395 SJ25C1 (CD19), and Brilliant Ultraviolet 737 B-ly4 (CD21). Flow cytometry and cell sorting were performed on BD FACS Aria II.

**Bulk antibody sequencing library generation and sequencing**. MIDs were added during the reverse transcription step through the use of fusion primers, which contain the partial Illumina P5 sequencing adaptor followed by twelve random nucleotides and primers to the constant region of five antibody isotypes. We fused eleven forward primers that were previously designed[6] to partial Illumina P7 adaptor. Full Illumina adaptors were added during the second PCR step along with library indexes. Total RNA was purified using All Prep DNA/RNA kit (Qiagen) following the manufacturer's protocol. cDNA synthesis was done using Superscript III (Life Technologies) with 30% of the RNA as input. After free primer removal by Exonuclease I digestion (New England Biolabs) according to the manufacturer's protocol, Takara Ex Taq HS polymerase (clone Tech) was used for both PCR reactions. The first PCR was performed with the following program: initial denature at 95°C for 3 min, followed by 20 cycles of 95°C for 30 s, 57°C for 30 s, and finally 72°C for 2 min with a 4°C hold. All cDNA was used for first PCR and about 1% of 1st PCR product was used for second PCR. The second PCR was performed with the following program: initial denature at 95°C for 3 min, followed by 10 cycles of 95°C for 30 s, 57°C for 30 s, and finally 72°C for 2 min with a 4°C hold. The second PCR products were gel purified and quantified by qPCR Library Quantification Kit (KAPA Biosystems, catalog number KK4824). 0.009 pmol of the QCed second PCR product were sequenced on Illumina Miseq with paired-end 250 bp read mode (sequencing kit catalog number: MiSeq Reagent Kit v2 (500cycle) MS-102-2003). The list of primers for RT and PCR can be found in Supplementary Table 1. Libraries were sequenced multiple times until saturated based on rarefaction analysis in Supplementary Fig. 4. Reads from all runs were combined and analyzed.

**Preliminary read processing**. Raw reads from Illumina MiSeq PE250 were first cleaned up following steps outlines in Fig. 1b. Only reads that exactly matched the corresponding library indices were included for further processing. The end of each raw read was trimmed such that all bases had a quality score of 25 or higher. Reads 1 and 2 were merged using the SeqPrep tool (https://github.comjstjohn/SeqPrep). The merged reads were filtered with specific V-gene and constant region primers to determine immunoglobulin (Ig)-sequencing reads. The primers were then

truncated from the reads. The retained reads were further truncated to 320 bp for the naive B cells in method verification experiments and 270 bp for samples from malaria cohort. Read numbers after each filter are listed in Supplementary Tables 2 and 4.

**MID sub-group generating**. Raw reads were split into MID groups according to their 12 nucleotide barcodes. For each MID group, quality threshold clustering was used to cluster similar reads. This process groups reads derived from a common template RNA molecule together, whereas separating reads derived from distinct RNA molecules. A Levenshtein distance of 15% of the read length was used as the threshold. This was calibrated using RNA controls with known sequences (Supplementary Fig. 1). For each sub-group, a consensus sequence was built based on the average nucleotide at each position, weighted by the quality score. In the case that there were only two reads in an MID sub-group, we only considered them useful reads if both were identical. Each MID sub-group is equivalent to an RNA molecule. Next, we merged all of the identical consensus to form unique consensus sequences, or unique RNA molecules, which were used to estimate the diversity and assess the sequencing depth in rarefaction analysis (Fig. 1c, d and Supplementary Fig. 4). Scripts for this section can be downloaded at https://github.com/utjianglab/MIDCIRS.

**Error rate calculation**. The difference between the consensus sequence for an RNA molecule and the raw reads associated with it represent the errors generated in either PCR or sequencing. The error rate can be calculated using the following formula (1):

$$\text{Error Rate(Raw)} = \frac{\sum_{i=1}^{N_\text{I}} \text{Diff}(i, \text{I})}{N_\text{I} \times L}, \tag{1}$$

where Diff (*i*,I) is the Levenshtein distance between read *i* and the consensus sequence in MID sub-group I, $N_\text{I}$ is the number of reads in MID sub-group I, and *L* is the read length.

To estimate the improved error rate using MIDCIRS, we equally divided the raw reads from one library into two data sets. The same MID sub-group generating process was performed on both data sets. By comparing the differences between the consensus sequences with identical MID between these two data sets, we can calculate the improved error rate for using MID sub-groups as:

$$\text{Error Rate(MID)} = \frac{\sum_{\text{I,J}} \text{Diff}(\text{I}, \text{J}) \times N_\text{I}}{\sum_\text{I} N_\text{I} \times L}, \tag{2}$$

where Diff(I,J) is the Levenshtein distance between the consensus I and consensus J, which have the identical MID, $N_\text{I}$ is the number of reads in MID sub-group I, and *L* is the read length.

**VDJ definition and mutation counts**. As described in previous work, similar methods were used to define the V, D, and J gene segments for all sequences[3, 5, 6]. From the International ImMunoGeneTics information system database (IMGT, http://www.imgt.org/textes/vquest/refseqh.html)[62], human heavy chain variable gene segment sequences (249 V-exon, 37 D-exon and 13 J-exon) were downloaded. Each unique sequence was first aligned to all 249V gene alleles. The specific V-allele with a maximum Smith–Waterman score was then assigned. In some cases, newly identified germline alleles, defined either by TIgGER[35] or our method (below), were added to the template sequences. J-segments and D-segments were then similarly assigned. The number of mutations from germline sequence was counted as the number of substitutions from the best aligned V and J templates as previously described[3, 5, 6]. The CDR3 was omitted due to the difficulty in determining the germline sequence. The germline sequences of V, D, and J gene segments were grouped by combining similar alleles into families using IMGT designation in VDJ correlation plots. In total, 58V, 27D, and 6J families were used.

**Novel allele detection**. To address the possibility of novel germline alleles inflating the observed number of mutations, new germline alleles were assembled. In short, IgM sequences for each individual were aligned and assigned to the traditional V-gene alleles in the IMGT database. If novel alleles exist in individuals, parts of unique RNA sequences will be assigned as mutations when they are actually derived from differences between novel and traditional alleles. The ratios of unmutated unique RNA molecules to those with one, two, three, and four mutations compared to the IMGT germline were determined, and if any were found to be less than 2 to 1, the alleles were flagged for further inspection. Unique RNA molecules were used to minimize the contributions of clonal expansion, and IgM sequences were used to minimize the contributions of SHM. Sequences within flagged alleles were then aligned to the closest IMGT germline to determine if the mutations were truly polymorphisms. When identical mutation patterns were observed in a minimum of 80% of all sequences in a flagged allele family, it was deemed a novel germline allele. For individuals with sorted naive B cells, novel alleles were generated from the naive B cell BCR sequences to complement those found in the bulk IgM sequences.

TIgGER was used as previously reported as another method to discover novel alleles[35]. TIgGER compares the mutation rate at a specific position to the overall number of mutations for sequences within the same assigned V-gene allele. Outliers within the low mutation region suggests the existence of a novel allele, and the shape of the curve can effectively distinguish between individuals homozygous and heterozygous for the novel allele.

Our new method and TIgGER have a 90% percent overlap in newly identified alleles. Discrepancies between the two methods were treated with a conservative estimation on the number of SHM, meaning we liberally included novel alleles as part of the germline gene segments. Non-overlapping novel alleles were manually inspected, and the union of novel alleles detected by TIgGER and our method was included as part of the germline gene segments. Sequences mapped to these novel alleles were excluded from our analysis, which accounts for an average 8% of all sequences (Supplementary Table 5).

**Translation from nucleotide to amino acid sequences.** Nucleotide sequences were translated into amino acid sequences based on codon translation as previously described[3, 5, 6]. The unique RNA sequences were inputted to IMGT High V quest to translate into amino acid sequences. The boundary of the CDR3 is defined by IMGT numbering for Ig and two conserved sequence markers of "Tyr-(Tyr/Phe)-Cys" to "Trp-Gly". CDR3 length was determined according to these anchor residues.

**Selection pressure.** The selection pressure was evaluated via BASELINe[37]. The unique RNA molecules of PBMC, populations were inputted to BASELINe and compared with the closest IMGT germline alleles. The observed number of replacement and silent mutations were compared with the expected number of mutations for the assigned germline sequence. A selection strength as measured by the probability density function (PDF) was generated using BASELINe to indicate the direction and degree for CDR (CDR1 and 2) and FWR (FWR2, and 3) regions for each unique RNA molecule first, then combined for each individual, and then further combined for each of the two groups, infants and toddlers. The PDFs were plotted and compared between infants and toddlers (Fig. 4). The associated P values comparing two group PDFs were calculated using BASELINe. Samples with 5,000,000 PBMCs were subsampled to 120,000 RNA molecules. Samples with fewer PBMCs were subsampled to proportionally fewer RNA molecules according to the PBMC number.

**Replacement/silent mutation.** According to the amino acid sequence translation results and V/D/J gene templates alignment results, we counted the number of nucleotide mutations resulting in amino acid substitutions (replacement, R) or no amino acid substitutions (silent, S) in FWR region (FWR2 and 3) and CDR region (CDR1 and 2) using the IMGT/HighV-QUEST[63]. The number of silent and replacement mutations was averaged in each age-group (infant and toddler) and the ratio for silent vs. replacement mutation was calculated. The CDR3 and FWR4 were omitted due to the difficulty in determining the germline sequence. FWR1 for all sequences was also omitted because it was not covered entirely by some of the primers. Samples with 5,000,000 PBMCs were subsampled to 120,000 RNA molecules. Samples with fewer PBMCs were subsampled to proportionally fewer RNA molecules according to the PBMC number.

**VDJ usage correlation.** The correlation of VDJ usage between infants and toddlers were calculated with Pearson correlation coefficient as the following formula (3) as in previous studies[3, 5, 6]:

$$corr = \frac{\sum_{v=\{V\},d=\{D\},j=\{J\}} \left(X_{vdj} - \langle X \rangle\right)\left(Y_{vdj} - \langle Y \rangle\right)}{\sqrt{\sum_{v=\{V\},d=\{D\},j=\{J\}} \left(X_{vdj} - \langle X \rangle\right)^2 * \sum_{v=\{V\},d=\{D\},j=\{J\}} \left(Y_{vdj} - \langle Y \rangle\right)^2}}, \quad (3)$$

$vdj$ refers to the combination of one $v$ allele family from 58V-gene allele families ($\{V\}$), one $d$ allele family from 27 D gene allele families ($\{D\}$), and one $j$ allele family from 6J gene allele families ($\{j\}$). For the reads weighted correlation, $X_{vdj}$ and $Y_{vdj}$ refer to the fraction of reads assigned to the respective $vdj$ combination for individuals X and Y, respectively. $\langle X \rangle$ and $\langle Y \rangle$ are the average reads across all $vdj$ combinations. For the lineage weighted correlation, these parameters refer to the fraction of lineages for each $vdj$ allele family combination. Samples with 5,000,000 PBMCs were subsampled to 120,000 RNA molecules. Samples with fewer PBMCs were subsampled to proportionally fewer RNA molecules according to the PBMC number. $vdj$ combinations that were not detected in either sample being compared were omitted from the correlation calculation.

**Clustering sequences into clonal lineages.** Sequences with similar CDR3 are possibly progenies from the same naive B cell and can be grouped into a clonal lineage. To detect the lineage structure for the antibody repertoire, we performed single linkage clustering, using a re-parameterization of the method described previously[5, 6, 44], accounting for the larger size of the CDR3 and junction in humans as compared to zebrafish. RNA sequences with the same V and J allele assignments, the same junction length, and whose junction regions differed by no

more than 10% on the nucleotide level were grouped together into a lineage. This is equivalent to a biological clone that underwent clonal expansion. To test the robustness of this threshold, we also tried the threshold of 95% similarity for CDR3 region[44], and it did not change the overall position of each lineage in the diversity-size plot, nor did it change the linear fits of lineage distribution generated from these two threshold (Supplementary Fig. 12). Lineage diversity is the number of unique RNA molecules within the lineage, and lineage size is the total number of RNA molecules within the lineage. Samples with 5,000,000 PBMCs were subsampled to 120,000 RNA molecules. Samples with fewer PBMCs were subsampled to proportionally fewer RNA molecules according to the PBMC number.

**Clonal lineage diversification.** In order to discuss the clonal lineage diversification, the size and diversity, as described above, were plotted against each other for pre-malaria and acute malaria time points for each individual (Fig. 5a). The linear regression visualizes the average degree of diversification relative to clonal expansion. A characteristic shift toward further diversification of clonal lineages upon acute malaria infection was evaluated by the decrease in the slope of the linear regression for each infant and toddler. The shift was calculated by the difference between the arctangents of the slopes of the linear regressions.

**Two time point-shared lineage analysis.** To test the effects of acute malaria infection on the structure of clonal lineages, RNA molecules from both the pre-malaria and acute malaria time points were grouped together and subjected to clustering into clonal lineages as described above. Resulting lineages that contained sequences from both the pre-malaria and acute malaria time points were isolated for mutational analysis. Within these shared lineages, the average number of mutations for the pre-malaria sequences was calculated alongside the average number of mutations for the acute malaria sequences (Fig. 6a).

**Lineage structure visualization.** Representative lineages were selected to visualize the lineage structures and the evolution of antibody sequences. Lineage structures were generated using COLT[52] (software can be downloaded here: http://www.cs.wright.edu/~keke.chen/software/colt.zip) and validated manually. We implemented a lineage visualization tool, COLT-Viz (He et al., submitted). In short, COLT considers constraints (e.g., isotype and timepoint) along with mutational patterns to build lineage trees. The height of each node is proportional to the number of RNA molecules associated with the unique sequence (size), the *color* of each node relates to the number of SHMs, and the distance between nodes is proportional to the Levenshtein distance between the node sequences.

**Pre-malaria memory B cells with acute progeny lineage analysis.** To determine the fate of the pre-malaria memory B cells upon acute malaria infection, two time point-shared lineages were formed as described above, and lineages containing sequences from both FACS-sorted pre-malaria memory B cells and acute malaria PBMCs were isolated for further analysis. COLT was used to generate lineage tree structures. Pre-malaria memory B cells that served as parent nodes to acute malaria sequences, as exemplified (Supplementary Fig. 15), were considered "pre-malaria memory B cells with acute progeny" (Fig. 6c–f).

**Code availability.** References or links for all software tools used are listed in the relevant "Methods" sections. All other relevant data are available from the authors.

**Data availability.** Sequencing data that support the findings of this study have been deposited in dbGaP with the accession code phs001209.v1.p1.

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

## Acknowledgements

The authors would like to thank WE ARE BLOOD, blood center, for providing the blood samples for method validation; Dr. Scott Hunicke-Smith, Jessica Podnar, and Dr. Michael Wilson at the Genomic Sequencing and Analysis Facility at UT Austin for helping with the sequencing runs; Dr. Evan Cohen, for helping with the cell sorting; Dr. Purnamrita Sarkar at the Department of Statistics and Data Sciences at UT Austin for advices on statistical testing. Dr. Steven Kleinstein, Jason Vander Heiden, and Dr. Gur Yaari for sending us an updated version of the BASELINe software and advices on data analysis. This work was supported by NIH Grants R00AG040149 (N.J.), S10OD020072 (N.J.), and R01GM106137 (P.R.) and by the Welch Foundation Grant F1785 (N.J.). N.J. is a Cancer Prevention and Research Institute of Texas (CPRIT) Scholar and a Damon Runyon-Rachleff Innovator. B.S.W. is a recipient of the Thrust 2000–George Sawyer Endowed Graduate Fellowship in Engineering. The cohort study in Mali was supported by the Division of Intramural Research, National Institute of Allergy and Infectious Diseases, National Institutes of Health.

## Author contributions

B.S.W. performed all malaria related experiment and data analysis; C.H. performed BASELINe and other sequencing data analysis; M.Q. developed the sequencing protocol using sorted naive B cells; D.W. helped with sequence analysis; S.M.H. helped with library construction; K.-Y.M. helped with sequencing; J.X. helped with lineage visualization, E.W.L, P.D.C., and S.K.P. selected malaria patients, provided samples and helped with experimental design; P.R. provided computation resources and helped with analysis; K.C. helped with lineage structure algorithm optimization and lineage visualization; N.J. conceived the idea, designed the study, and directed data analysis; B.S.W. and N.J. wrote the paper with contributions from all co-authors.

## Additional information

**Competing interests:** The authors declare that N.J. is a scientific advisor of ImmuDX, LLC and a provisional patent application has been filed by the University of Texas at Austin for the method described here.

