## [Peer Review File · Nature Communications]

Reviewers' comments:

Reviewer #2 (Remarks to the Author):

Previous critiques have been addressed

Reviewer #3 (Remarks to the Author):

This is a second revision of the manuscript that I review. The authors build the manuscript around MIDCIRS, "a highly accurate and high-coverage repertoire sequencing method, which can be applied to as few as 1,000 naïve B cells (NaiBs)". As such I believe that sufficient details should be given so that other labs could apply this method. Currently, there is not sufficient information for doing this. More details are needed in the following steps:

Data analysis:

The authors say: "All codes used will be provided upon request to the corresponding author." IMHO detailed scripts should be publicly available without the need to email the corresponding author. Scripts should contain version information about third party software tools to allow for reproducing the results presented in the paper.

Cell sorting:

1. Details on how the cells were treated are missing. For example, did the authors perform FACS sorting immediately after thawing the PBMCs?
2. Which FACS sorter was used?
3. What was the cell surface Immunofluorescence staining protocol?
4. Were the cells lysed immediately after the FACS sorting without recovery?
5. What were the fluorophores on the antibodies? Or what was the cat. number for each antibody.

Bulk antibody sequencing library generation and sequencing:

1. For the cDNA synthesis, how many cells were used? Or how much RNA (in ng), whichever was used for normalization.
2. How did the authors remove free primers? And did they perform removal after each PCR step?
3. What percent of cDNA was taken for PCR1?
4. What percent of PCR1 was taken for PCR2?
5. What were the cat. numbers for the qPCR Library Quantification Kit (KAPA biosystems) and for the MiSeq Reagent Kit?
6. How much DNA was loaded on the MiSeq (pm)?
7. How much PhiX was used?

Point-by-point responses to reviewers' comments.

Reviewers' comments:

Reviewer #2 (Remarks to the Author):

Previous critiques have been addressed

Reviewer #3 (Remarks to the Author):

This is a second revision of the manuscript that I review. The authors build the manuscript around MIDCIRS, “a highly accurate and high-coverage repertoire sequencing method, which can be applied to as few as 1,000 naïve B cells (NaiBs)”. As such I believe that sufficient details should be given so that other labs could apply this method. Currently, there is not sufficient information for doing this. More details are needed in the following steps:

We thank reviewer 3 for these comments. Detailed information on informatics pipeline and experimental protocol have been added to the revised manuscript with changes highlighted.

Data analysis:

The authors say: “All codes used will be provided upon request to the corresponding author.” IMHO detailed scripts should be publicly available without the need to email the corresponding author. Scripts should contain version information about third party software tools to allow for reproducing the results presented in the paper.

References or links for all software tools used are listed in their respective supplementary materials sections.

Cell sorting:

1. Details on how the cells were treated are missing. For example, did the authors perform FACS sorting immediately after thawing the PBMCs?

PBMCs were thawed, washed with RPMI, and then processed for FACS. These details have been added to the Supplementary Methods under the subheading “Cell sorting”.

2. Which FACS sorter was used?

BD FACS Aria II was used for flow cytometry and cell sorting. This has been added to the Supplementary Methods under the subheading “Cell sorting”.

3. What was the cell surface Immunofluorescence staining protocol?

Standard cell surface immunofluorescence staining protocol was followed, which was also used in our previous studies. This has been clarified in the Supplementary Methods under the subheading “Cell sorting”.

4. Were the cells lysed immediately after the FACS sorting without recovery?

Yes, cells were immediately lysed after FACS sorting.

5. What were the fluorophores on the antibodies? Or what was the cat. number for each antibody.

Fluorophore information was added to the Supplementary Methods under the subheading “Cell sorting”.

Bulk antibody sequencing library generation and sequencing:

1. For the cDNA synthesis, how many cells were used? Or how much RNA (in ng), whichever was used for normalization.

Cell numbers are indicated in supplementary tables 2 (naïve-sorted controls), 4 (malaria PBMCs), 6 (malaria naïve-sorted), and 8 (malaria MemB-sorted, table caption). 30% of purified RNA was used for cDNA synthesis, as stated in the Supplementary Methods under the subheading “Bulk antibody sequencing library generation and sequencing”. Normalization was performed according to cell count, as indicated in the Supplementary Methods under the relevant subsections.

2. How did the authors remove free primers? And did they perform removal after each PCR step?

Exonuclease I (New England Biolabs) was used for digestion of free RT primers according to the manufacturer’s protocol. These details were added to the Supplementary Methods under the subheading “Bulk antibody sequencing library generation and sequencing”. Primer removal was not performed after PCR steps.

3. What percent of cDNA was taken for PCR1?

All cDNA was used for PCR1. This has been added to the Supplementary Methods under the subheading “Bulk antibody sequencing library generation and sequencing”.

4. What percent of PCR1 was taken for PCR2?

About 1% of 1st PCR product was used for second PCR. This has been added to the Supplementary Methods under the subheading “Bulk antibody sequencing library generation and sequencing”.

5. What were the cat. numbers for the qPCR Library Quantification Kit (KAPA biosystems) and for the MiSeq Reagent Kit?

KAPA biosystems, catalog number KK4824 and MiSeq Reagent Kit v2 (500cycle) MS-102-2003 were used. These details have been added to the Supplementary Methods under the subheading “Bulk antibody sequencing library generation and sequencing”.

6. How much DNA was loaded on the MiSeq (pm)?

0.009pmol. This has been added to the Supplementary Methods under the subheading “Bulk antibody sequencing library generation and sequencing”.

7. How much PhiX was used?

It is not a standard operation for most of the sequencing cores to add PhiX to Mi-seq runs. Thus, PhiX was not added. All our runs passed Illumina QC and more than 85% of bases with quality score > Q30, which is much higher than Illumina specification of 75%.

REVIEWERS' COMMENTS:

Reviewer #3 (Remarks to the Author):

Previous critiques have been addressed